# Defects in integrin complex formation promote *CHKB*-mediated muscular dystrophy

Mahtab Tavasoli , Christopher R McMaster

Phosphatidylcholine (PC) is the major membrane phospholipid in most eukaryotic cells. Bi-allelic loss of function variants in *CHKB*, encoding the first step in the synthesis of PC, is the cause of a rostrocaudal muscular dystrophy in both humans and mice. Loss of sarcolemma integrity is a hallmark of muscular dystrophies; however, how this occurs in the absence of choline kinase function is not known. We determine that in $Chkb^{-/-}$ mice there is a failure of the $\alpha7\beta1$ integrin complex that is specific to affected muscle. We observed that in $Chkb^{-/-}$ hindlimb muscles there is a decrease in sarcolemma association/abundance of the $PI(4,5)P_2$ binding integrin complex proteins vinculin, and $\alpha$-actinin, and a decrease in actin association with the sarcolemma. In cells, pharmacological inhibition of choline kinase activity results in internalization of a fluorescent $PI(4,5)P_2$ reporter from discrete plasma membrane clusters at the cell surface membrane to cytosol, this corresponds with a decreased vinculin localization at plasma membrane focal adhesions that was rescued by overexpression of *CHKB*.

## Introduction

The specialized plasma membrane of skeletal muscle, called sarcolemma, transmits force and is subjected to substantial stress during muscle contraction (1). To transmit and withstand force, the sarcolemma is firmly attached to the ECM via sub-membranous Z-line associated structures called costameres. The costamere is akin to the more well-known focal adhesion complex present in most cells and consist of two major protein complexes: the dystrophin-glycoprotein complex (DGC) and the integrin-vinculin-talin complex (2, 3). The costamere is a critical structural anchor for the sarcomere by providing support during muscle contraction. The importance of proper costamere function in health and disease is apparent as of the over 30 genes, where variants can cause muscular dystrophy; many are in genes encoding DGC complex proteins (4, 5, 6, 7).

A thought-provoking form of muscular dystrophy is because of autosomal recessive loss of function of the *CHKB* gene (#602541;

OMIM) (8, 9, 10). *CHKB*-mediated muscular dystrophy is the only defect in the synthesis of a major membrane lipid known to cause a muscular dystrophy (9, 10, 11, 12, 13). *CHKB* encodes choline kinase β, the first enzymatic step in the synthesis of phosphatidylcholine (PC), the most abundant phospholipid in eukaryotic membranes comprising 40–60% of total phospholipid in most eukaryotic cell types (14, 15, 16). Human (and mouse) contain a second choline kinase isoform encoded by the *CHKA* gene, and together with *CHKB*, provide the total choline kinase capacity for PC synthesis via the Kennedy pathway (15, 16). In mice, loss of *CHKB* function results in a muscular dystrophy that present a rostrocaudal gradient with proximal muscle most affected (10, 17). Interestingly expression of the *Chka* gene in mice modulates the muscular dystrophy phenotype of its ortholog *Chkb* as (i) in $Chkb^{-/-}$ mice the affected proximal muscles show a decrease in Chka protein level, whereas unaffected distal muscles show an increase Chka level, and (ii) viral-mediated expression of *Chka* in affected muscle of $Chkb^{-/-}$ mice ameliorated the muscular dystrophy phenotype (17, 18, 19, 20). Thus, near complete loss of function of choline kinase in muscle is necessary for the muscular dystrophy to arise. Interestingly the level of the product, PC, of the Kennedy pathway is unchanged in affected muscle in *CHKB* muscular dystrophy patients and in $Chkb^{-/-}$ mice as there is in increase in PC uptake from serum (17, 19). Instead a plethora of lipid metabolic defects occur because of an inability of cells to consume a downstream substrate of this pathway, diacylglycerol (DG) and the upstream substrates that produce it. The result is a temporal change in lipid metabolism in affected muscle beginning with an inability to consume fatty acids for use as energy in the early stage of the disease followed by shunting of fatty acids into triacylglycerol (TG) and an accumulation of lipid droplets in affected muscle as the disease progresses, providing yet another example of the integrated metabolic circuitry regulating lipid metabolism (19, 21, 22). If there are changes in other lipids, and if these affect the etiology of *CHKB*-mediated muscular dystrophy, have yet to be determined.

Loss of plasma membrane integrity in affected muscle is a hallmark of muscular dystrophies, and consistent with this an increase in creatine kinase level is observed in plasma of *CHKB* patients and $Chkb^{-/-}$ mice (6, 9, 11). It is well known that membrane lipid composition and localization play an important role in the maintenance of membrane integrity, however, as the level of the

Department of Pharmacology, Dalhousie University, Halifax, Canada

Correspondence: christopher.mcmaster@dal.ca; tavasoli@dal.ca

major membrane phospholipid PC was unchanged in affected muscle in $Chkb^{-/-}$ mice, it is unclear how membrane integrity is breached (10, 17). Interestingly, the DGC complex has been demonstrated to be intact in affected muscle in $Chkb^{-/-}$ mice; thus, how loss of choline kinase activity causes muscle cells lose integrity is unclear (10, 17). In this study, we focused on the integrin-vinculin-talin complex as many of its components are amphipathic proteins that exist in inactive soluble forms and are recruited to the plasma membrane to ensure proper costamere formation. The major recruiter of the amphipathic proteins of the integrin-vinculin-talin complex to the plasma membrane is phosphatidylinositol-4,5-bisphosphate (PI(4,5)P$_2$) (23, 24, 25, 26, 27, 28). PI(4,5)P$_2$ is a minor lipid present as less than 1% of total phospholipid in cells, but it is highly enriched on the inner leaflet of the plasma membrane (26). PI(4,5)P$_2$ binding for plasma membrane binding and localization by vinculin, talin, and $\alpha$-actinin are all necessary for proper function of the components of the integrin-vinculin-talin complex at the costamere (24, 25, 26, 29, 30, 31, 32, 33). Specifically, PI(4,5)P$_2$ binding by vinculin results in unmasking of binding sites for the integrin-vinculin-talin costamere proteins $\alpha$-actinin, talin, and actin (23, 24, 25, 29). Mutants of vinculin deficient in PI(4,5)P$_2$ binding are nonfunctional (3, 34). PI(4,5)P$_2$ also binds $\alpha$-actinin and PI(4,5)P$_2$ binding is thought to be required for $\alpha$-actinin to bind and cross-link actin (29, 30, 31, 35). The third PI(4,5)P$_2$ binding protein within the integrin-vinculin-talin complex is talin. The binding of vinculin to talin, and vice versa, is thought to be required for function of the integrin-vinculin-talin complex (32, 36). Talin provides the direct link to the ERM via its binding to integrins. PI(4,5)P$_2$ binding by talin is believed to be necessary for integrin clustering, which in turn mechanically connects this costamere complex to the ECM to facilitate muscle contraction (23, 27, 32, 33, 36, 37).

In this study, we present evidence that, specific only to affected muscle of $Chkb^{-/-}$ mice, there is decreased affinity of the integrin-vinculin-talin complex PI(4,5)P$_2$ binding proteins vinculin and $\alpha$-actinin for the plasma membrane resulting in disruption of integrin-mediated linkage between actin filaments and sarcolemma. Consistent with this observation, a decrease in choline kinase activity results in a reduction in the level of a PI(4,5)P$_2$ probe at the plasma membrane and its accumulation in the cytosol concomitant with vinculin redistribution, with both the PI(4,5)P$_2$ probe and vinculin returning to the plasma membrane upon overexpression of Chkb.

## Results and Discussion

### Decreased integrin-mediated linkage between cytoskeleton and ECM in affected muscle from Chkb-deficient mice

$CHKB$-mediated muscular dystrophy in $Chkb^{-/-}$ mice shows a rostrocaudal gradient of injury with proximal muscles being most affected. We used transmission electron microscopy to examine the overt structure of muscle in the forelimb (unaffected) and hindlimb (affected) in 12-d old WT and $Chkb^{-/-}$ mice. Compared with WT mice, the $Chkb^{-/-}$ mice show extensive sarcomere degeneration in hindlimb (quadriceps and gastrocnemius) but not forelimb (triceps) (Fig 1A). In the $Chkb^{-/-}$ hindlimb both the Z line

and I bands are absent in some areas and the myofibrils are narrow and split. The Z line forms the periphery of a sarcomere where actin filaments attach, whereas the I band is the area around the Z line consisting mainly of thin actin filaments (38). The defect in Z line and I band formation only in the affected muscle of $Chkb^{-/-}$ mice is consistent with a defect in costamere function.

As it was previously demonstrated that there was no defect in the DGC complex of the costamere in either forelimb or hindlimb muscle of $Chkb^{-/-}$ mice (10), we turned our attention to the integrin-vinculin-talin complex. We analyzed the amount of vinculin at the membrane by confocal microscopy using an anti-vinculin antibody. Actin was also stained using Alexa Fluor 488-phalloidin. As expected, in WT mice vinculin was present at the plasma membrane in hindlimb muscle cells, whereas in $Chkb^{-/-}$ mice, vinculin was mislocalized (Fig 1B). Actin staining was also disturbed in the hindlimb of $Chkb^{-/-}$ mice consistent with the defect in Z line and I band structures we observed by electron microscopy in the hindlimb of $Chkb^{-/-}$ mice. We further analyzed the amount of vinculin in the membrane/cytoskeletal fraction of hindlimb muscle in wild-type and $Chkb^{-/-}$ mice by treatment with cytoskeleton stabilization buffer (CSK). CSK treatment removes proteins that are loosely attached to the membrane/cytoskeleton from cells, whereas proteins that bind tightly to the membrane/cytoskeleton are not solubilized and remain on the substrate (39, 40, 41). As expected, in wild-type mice CSK treatment did not result in loss of vinculin from the plasma membrane of hindlimb muscle, whereas in $Chkb^{-/-}$ mice CSK treatment resulted in vinculin being washed out from these cells (Fig 1B and D). Quantification of vinculin immunostaining showed that there is a modest increase in total vinculin level $Chkb^{-/-}$ mice compared with wild-type mice, with the vinculin in $Chkb^{-/-}$ mice being internalized and poorly associated with the plasma membrane/cytoskeleton (Fig 1C and E). Overt differences in costamere formation in affected muscle of $Chkb^{-/-}$ mice are accompanied by a loss of plasma membrane/cytoskeleton association of the amphipathic integrin-vinculin-talin complex protein vinculin. This implies that the integrin-vinculin-talin complex may not be intact in affected muscle of $Chkb^{-/-}$ mice.

The observation that vinculin in hindlimb muscle of $Chkb^{-/-}$ mice is no longer membrane associated led us to further examine the main components of the integrin-vinculin-talin complex in skeletal muscle of $Chkb^{-/-}$ mice. We used Western blot to assess the protein levels of vinculin, metavinculin (a muscle-specific splice variant characterized by a 68-amino acid insert within the C-terminal tail domain), Itga7, talin, and $\alpha$-actinin. The level of each of these components was similar in WT and $Chkb$-deficient mice in forelimb (unaffected) muscle (Fig 2A). In $Chkb$-deficient hindlimb muscle there was a strong increase in total vinculin, Itga7 and talin and a significant decrease in the levels of $\alpha$-actinin and metavinculin (Fig 2B). CSK treatment determined that much less vinculin, $\beta$-actin and $\alpha$-actinin was associated with the membrane/cytoskeleton in $Chkb^{-/-}$ hindlimb muscle compared with $Chkb^{+/+}$ mice (Fig 3A–C). It is clear there is a discordance in the abundance of proteins that comprise the integrin-vinculin-talin complex and a decrease in their association with membrane/cytoskeleton in affected muscle in $Chkb^{-/-}$ mice.

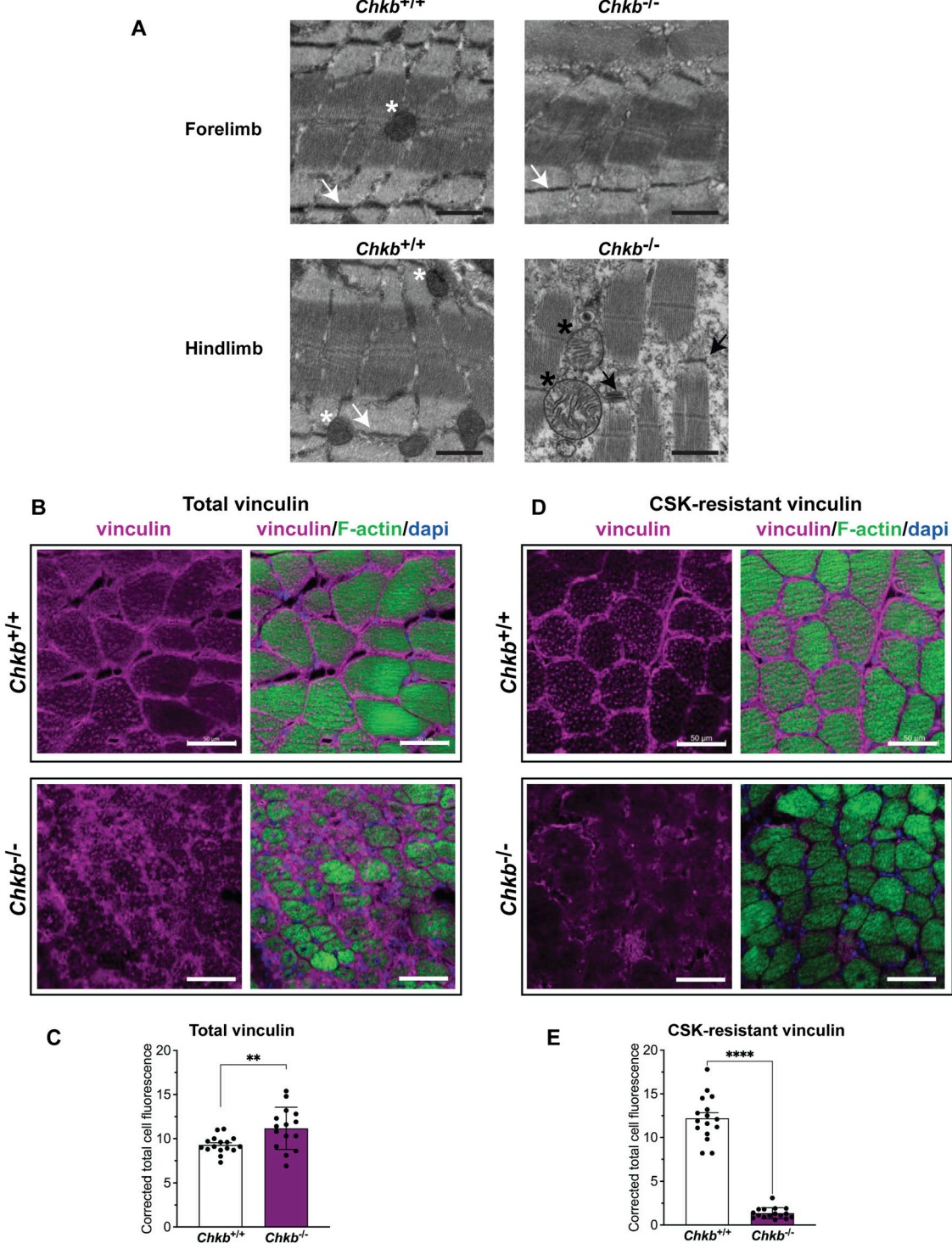

**Figure 1. Decreased cytoskeleton-associated vinculin in hindlimb muscles from Chkb-deficient mice.**
**(A)** Transmission electron microscopy of the forelimb and hindlimb from 12-d-old wild-type ($Chkb^{+/+}$) and Chkb-deficient ($Chkb^{-/-}$) mice shows aligned Z-bands (white arrows) and normal mitochondria (white asterisk) in hindlimb and forelimb muscle samples from wild-type and forelimb samples from $Chkb^{-/-}$ mice. Disrupted integrity of intercalated disk and Z-line (black arrow), mitochondrial degeneration with erased cristae (black asterisk), and prominent sarcomere degeneration in hindlimb samples from $Chkb^{-/-}$ mice. Scale bar = 800 nm. **(B, C, D, E)** Decreased cytoskeletal association of vinculin in hindlimb muscles from $Chkb^{-/-}$ mice compared with their wild-type littermates. **(B, D)** Representative micrograph of frozen hindlimb skeletal muscle tissue from wild-type and $Chkb$-deficient mice fixed without (B) or with (D) cytoskeleton stabilization buffer (CSK) treatment. Tissue sections were immuno-stained using anti-vinculin (purple) and Alexa Fluor 488-phalloidin (actin, green) antibody. Images were obtained using confocal microscopy. CSK treatment removes cytoplasmic proteins as well as proteins that are loosely attached to the cytoskeleton from cells but not proteins tightly binding to cytoskeleton. Thus, activated and cytoskeleton-associated vinculin can be visualized by CSK treatment. There is an increase in total vinculin

### Reduced choline kinase activity results in redistribution of PI(4,5)P₂ from the plasma membrane and decreased vinculin localization in focal adhesions

The binding of vinculin, talin, and $\alpha$-actinin to PI(4,5)P$_2$ at the plasma membrane is required for integrin complex formation at both costameres and focal adhesion sites (23, 24, 25, 26, 27, 29, 30, 31, 34). Phospholipid metabolism is highly integrated as demonstrated by our previous work demonstrating alterations the levels of many lipids in a temporal manner in affected muscle in $Chkb^{-/-}$ mice (19, 21). This previous work also determined that in $Chkb^{-/-}$ mouse forelimb or hindlimb muscle, Chkb protein expression is undetectable, whereas in forelimb muscle from $Chkb^{-/-}$ mice there is a compensatory up-regulation of Chka protein expression to almost threefold that observed in WT mice, whereas in hindlimb muscle from $Chkb^{-/-}$ mice Chka protein expression is decreased to less than 10% that observed in WT mice (19). Thus, a near total loss of choline kinase activity is a major driver of the changes in lipid metabolism and the etiology of the muscle defects observed in $Chkb^{-/-}$ mice. As components of the integrin-vinculin-talin complex bind membranes/cytoskeleton in a PI(4,5)P$_2$ dependent manner and their level and/or association with membranes/cytoskeleton is decreased in affected muscle of $Chkb^{-/-}$ mice, we sought to determine if PI(4,5)P$_2$ localization was affected because of decreased choline kinase activity. To do so, we treated U2O2 cells with two different choline kinase inhibitors, EB-3D and CK-37, and used the well characterized PI(4,5)P$_2$ localization reporter PH-PLCD1-GFP (42, 43).

Confocal microscopy revealed that in control cells dorsal plasma membrane clusters of PH-PLCD1-GFP were observed and these decreased in U2OS cells treated with either choline kinase inhibitor (Fig 4A–C). Consistent with a decrease in the plasma membrane localization of the PI(4,5)P$_2$ reporter upon inhibition of choline kinase activity, the PI(4,5)P$_2$ binding costamere protein vinculin was detached from the plasma membrane because of choline kinase inhibitor treatment (Fig 4D and E). To ensure that this relocalization was solely because of inhibition of choline kinase activity, in the presence of either choline kinase inhibitor overexpression of $CHKB$ (Fig 4F) resulted in localization of the PI(4,5)P$_2$ reporter back to the plasma membrane (Fig 4G–I) concomitant with vinculin translocation to the plasma membrane (Fig 4J and K). A decrease in choline kinase activity results in a decrease in the level of the PI(4,5)P$_2$ reporter from plasma membrane and decreased affinity of the PI(4,5)P$_2$ binding costamere component vinculin for the plasma membrane.

To determine if plasma membrane phospholipid redistribution from the plasma membrane was a general phenomenon when choline kinase activity was decreased we also determined the localization of phosphatidylserine (PS). Like PI(4,5)P$_2$, PS is highly enriched in inner leaflet of the plasma membrane, with PS also found in late endocytic compartments and recycling endosomes (44, 45, 46, 47). To determine PS localization we used a well

characterized PS binding probe consisting of the C2 domain of lactadherin (LactC2) linked with monomeric RFP (mRFP-Lact-C2). As previously shown, the mRFP-Lact-C2 PS labeled the plasma membrane of U2OS cells and internal structures that have previously been determined to be endosomes (Fig 5A) (48). Next we determined if choline kinase inhibition altered PS distribution, using redistribution of the PI(4,5)P$_2$ probe as our control. Choline kinase inhibition did not change the distribution of the PS probe and, as expected, did result in internalization of the PI(4,5)P$_2$ probe. Interestingly, in the confocal images the internalized PI(4,5)P$_2$ probe appeared to colocalize with the cytosolic portion of the PS probe (Fig 5A–C). We quantified the extent of this PI(4,5)P$_2$ and PS colocalization using both the Manders' correlation coefficient and Pearson's correlation coefficient in vehicle-treated and choline kinase inhibited cells (49). Choline kinase inhibition increased the fraction of PI(4,5)P$_2$ overlapping PS (Fig 5D), and the fraction of PS overlapping PI(4,5)P$_2$ (Fig 5E), as well as the Pearson's correlation coefficient of images of PH-PLCD1-GFP and PS reporter mRFP-Lact-C2 in U2OS cells (Fig 5F). As PS distribution did not change upon choline kinase inhibition this suggests that (i) PS localization is not dependent on PC synthesis via the Kennedy pathway and (ii) there are no large overt changes in the PM upon choline kinase inhibition suggesting that the plasma membrane itself is not affected in an overt manner. These observations are consistent with internalization of the PI(4,5)P$_2$ probe to endosomes, but do not rule out that the decrease in PI(4,5)P$_2$ probe signal at the plasma membrane could also be because of decreased synthesis or increased catabolism of PI(4,5)P$_2$ itself. This will be a line of interesting research to pursue.

Loss of plasma membrane/ECM integrity in muscle is a hallmark of muscular dystrophies including $CHKB$-mediated muscular dystrophy. Our work provides a rationale for this phenomenon in $CHKB$ patients and $Chkb^{-/-}$ mice and has revealed a previously unknown link between synthesis of the major membrane phospholipid PC and PI(4,5)P$_2$ presence at the plasma membrane. The fact that loss of function gene variants in $CHKB$, encoding the first step in the synthesis of PC, causes a muscular dystrophy enabled us to put this into a disease context. A decrease in choline kinase activity results in less PI(4,5)P$_2$ at the plasma membrane resulting in the PI(4,5)P$_2$ binding components of the integrin-vinculin-talin complex no longer binding to the plasma membrane/ECM. Loss of integrin-vinculin-talin complex integrity is a likely contributor to the decrease in muscle cell integrity in affected muscle in $CHKB$-mediated muscular dystrophy.

## Materials and Methods

### Animal genotyping and animal ethics

All animal procedures were approved by the Dalhousie University's Committee on laboratory animals in accordance with guidelines of

---

levels and much less cytoskeleton-associated vinculin in hindlimb skeletal muscle samples from $Chkb^{-/-}$ mice compared with the wild type. (Representative of three mice per group). Scale bar = 50 $\mu$m. **(C, E)** The corrected total cell fluorescence intensity of vinculin in hindlimb muscles from $Chkb^{+/+}$ and $Chkb^{-/-}$ mice fixed without (C) or with (E) CSK treatment. For (C, E), a total of 20 random myofibers were quantified per group in three distinct mice. Data are mean ± SD. **$P < 0.01$. $t$ test.

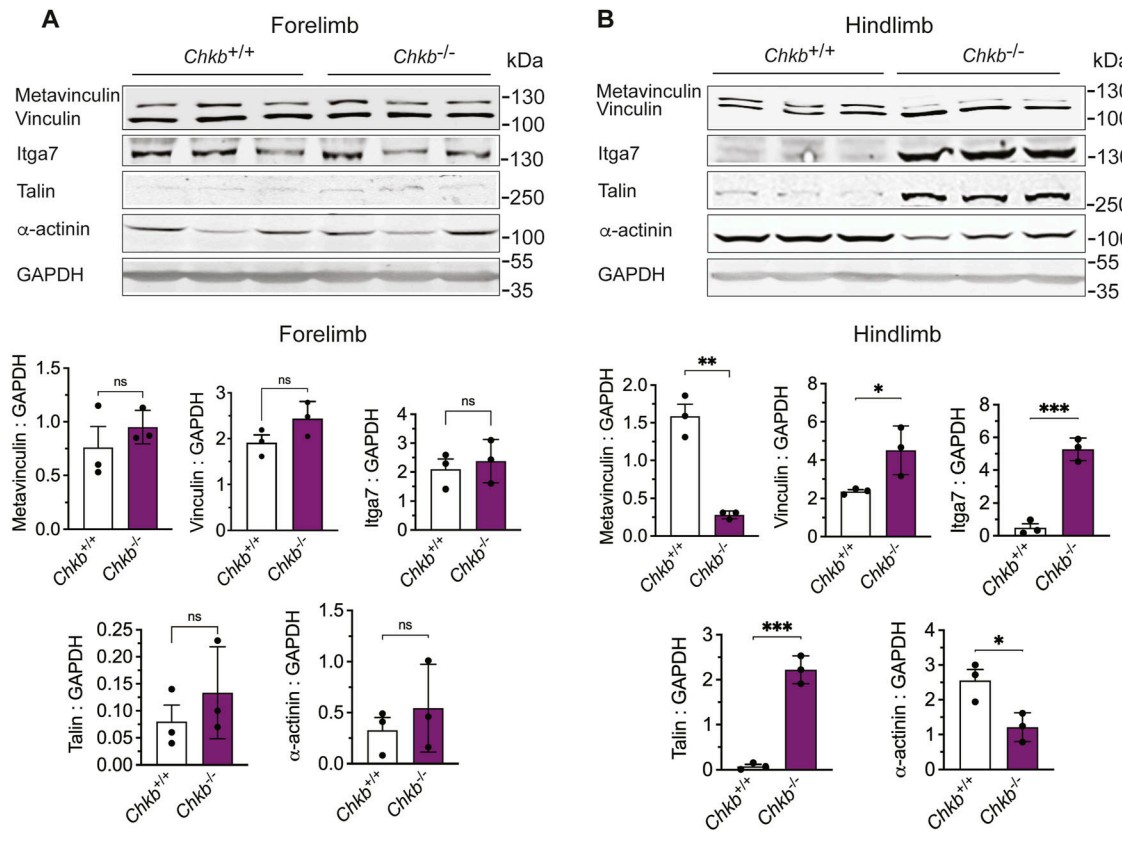

**Figure 2. Decreased abundance and sarcolemma association of integrin-mediated linkage components in affected muscle of *Chkb*⁻/⁻ mice.**
(A, B) Western blot of total tissue lysis of (A) forelimb (triceps) and (B) hindlimb (quadriceps) samples from three distinct (lanes 1–3) *Chkb*⁺/⁺ and three distinct (lanes 4–6) *Chkb*⁻/⁻ mice probed with anti-vinculin, anti-itga7, anti-talin, anti-α-actinin, and anti-Gapdh antibodies. Bottom: densitometry of the Western blot data show the ratio of vinculin, metavinculin, itga7, talin, and a-actinin to Gapdh in forelimb (triceps) and hindlimb (quadriceps) samples. Values are means ± SD; n = 3–4 per group. *P < 0.05, **P < 0.01, ***P < 0.001 (t test). Detergent (NP40), soluble (cytosolic).

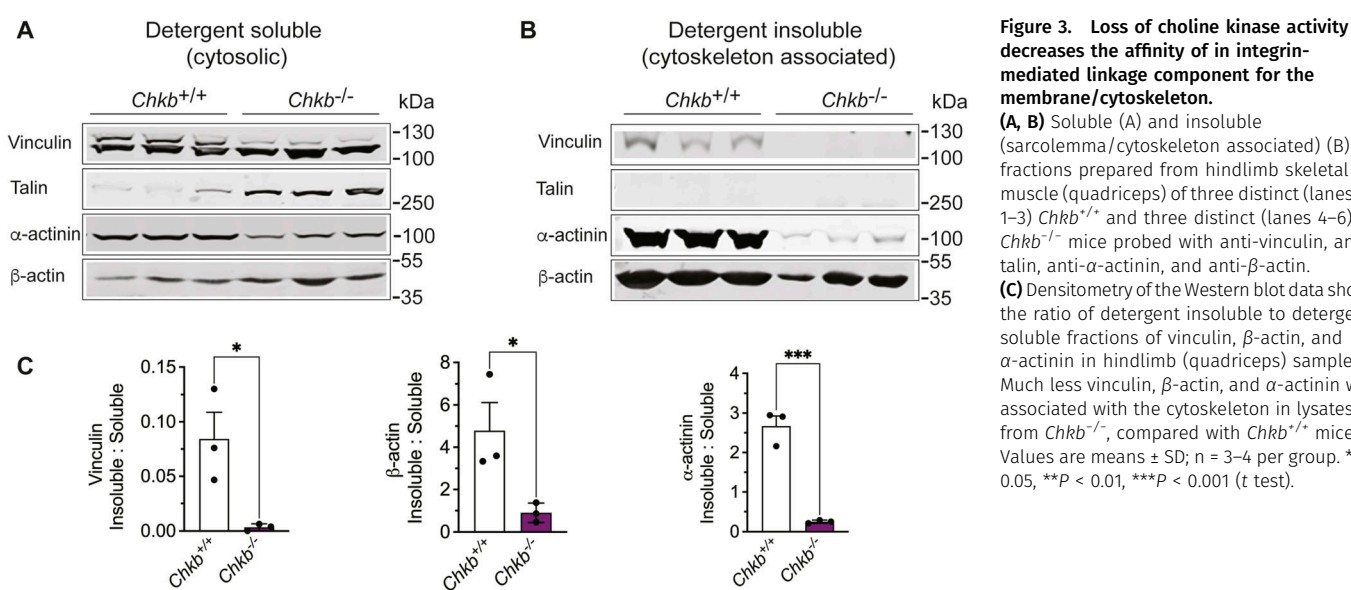

**Figure 3. Loss of choline kinase activity decreases the affinity of in integrin-mediated linkage component for the membrane/cytoskeleton.**
(A, B) Soluble (A) and insoluble (sarcolemma/cytoskeleton associated) (B) fractions prepared from hindlimb skeletal muscle (quadriceps) of three distinct (lanes 1–3) *Chkb*⁺/⁺ mice and three distinct (lanes 4–6) *Chkb*⁻/⁻ mice probed with anti-vinculin, anti-talin, anti-α-actinin, and anti-β-actin antibodies.
(C) Densitometry of the Western blot data show the ratio of detergent insoluble to detergent soluble fractions of vinculin, β-actin, and α-actinin in hindlimb (quadriceps) samples. Much less vinculin, β-actin, and α-actinin was associated with the cytoskeleton in lysates from *Chkb*⁻/⁻, compared with *Chkb*⁺/⁺ mice. Values are means ± SD; n = 3–4 per group. *P < 0.05, **P < 0.01, ***P < 0.001 (t test).

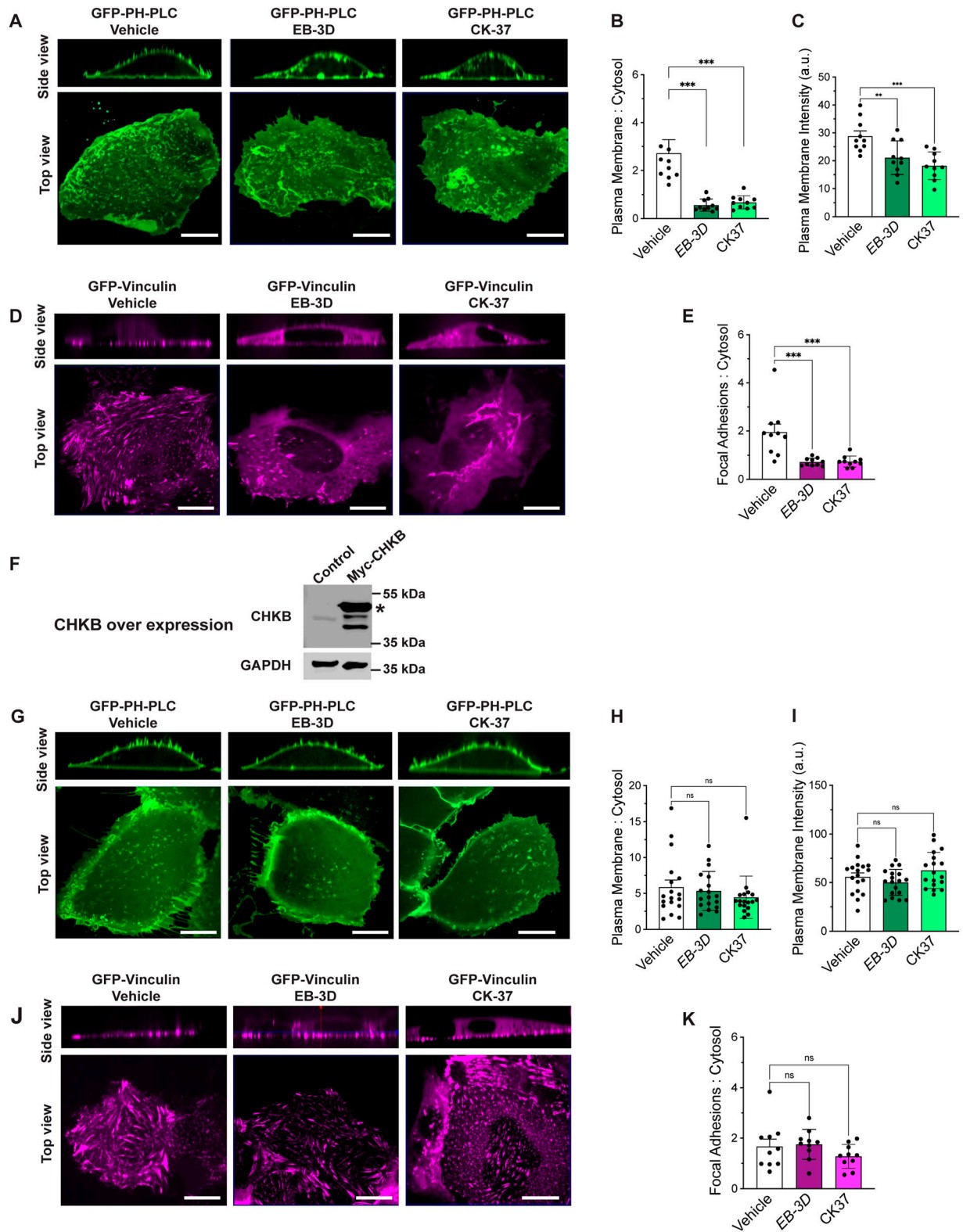

**Figure 4.   Treatment with choline kinase alpha inhibitors results in mis-localization of PI(4,5)P₂ and decreases vinculin localization in focal adhesions in control U2OS cells but has no effect on U2OS cells overexpressing Myc-CHKB.**

**(A)** U2OS cells were transiently transfected with the PI(4,5)P₂ reporter PH-PLCD1-GFP and were treated with the choline kinase alpha inhibitors EB-3D (10 mM), CK-37 (15 mM) or vehicle for 48 h and live cells were imaged by spinning-disk confocal microscopy. Choline kinase alpha inhibition resulted in the loss of dorsal membrane PH-PLCD1-GFP clusters in the U2OS cells. **(B)** The ratio of dorsal plasma membrane: cytoplasmic PH-PLCD1-GFP fluorescence is significantly decreased in choline kinase

the Canadian Council on Animal Care Guide to the Care and Use of Experimental Animals (CCAC: vol. 1, second ed., 1993; vol. 2, 1984). Chkb mice in the C57BL/6J background were the gift of Professor Gregory A. Cox and were from the Jackson Laboratory. Male $Chkb^{+/−}$ mice were crossed with female $Chkb^{+/−}$ mice to generate $Chkb^{+/+}$, $Chkb^{−/−}$, and $Chkb^{+/−}$ genotypes.

## Transmission electron microscopy

For TEM analysis, ~5 × 5 mm cubes of quadriceps, gastrocnemius and triceps were fixed with 2.5% Glutaraldehyde diluted with 0.1 M sodium cacodylate buffer and postfixed with 1% osmium tetroxide in Millonig's buffer solution for 2 h, dehydrated, and embedded in epon araldite resin. Ultrathin sections were stained with 2% uranyl acetate for 30 min and lead citrate for 4 min and viewed with a JEOL JEM 1230 Transmission Electron Microscope at 80 kV. Images were captured using a Hamamatsu ORCA-HR digital camera. Three mice per genotype for each timepoint were evaluated.

## Western blot analysis (WB) and quantification

Total protein extracts (T) were prepared as described previously (19) and subjected to SDS–PAGE. Briefly, the muscle tissue (~100 mg) was homogenized with a steel bead in 1 ml of cold RIPA buffer containing 1X Proteinase Inhibitor Mix (complete Protease Inhibitor Cocktail, Cat. no.11 697 498 001; Roche), 1X PhosStop (Mannheim Germany, Cat. no.04 906 845 001; Roche) using a TissueLyser II instrument (QIAGEN) set at 30 strokes/s for 2–4 min. Based on protein quantification results, all samples were adjusted to the final concentration of 2 µg/ul and heat-denatured for 5 min at 99°C in 2X Laemmli buffer. Proteins were separated by SDS–PAGE and transferred to nitrocellulose membranes. The membranes were incubated in Odyssey blocking solution for 1 h. Total proteins were detected by probing the membranes with appropriate primary antibodies overnight at 4°C. For intermediate filament-enriched cytoskeletal extracts, the muscle tissue (~100 mg) was homogenized with a steel bead in 1 ml of ice-cold low-salt buffer (10 mM Tris–HCl, pH 7.6, 140 mM NaCl, 5 mM EDTA, 5 mM EGTA, 0.5% Triton X-100, 2 mM phenylmethylsulfonyl fluoride), using a TissueLyser II instrument (QIAGEN) set at 30 strokes/s for 2–4 min. A subsequent centrifugation step was performed to separate the soluble cytoskeletal fraction (S) from the insoluble cytoskeletal fraction (C), which was then resuspended in ice-cold high-salt buffer (10 mM Tris–HCl, pH. 7.6, 140 mM NaCl, 1.5 M KCl, 5 mM EDTA, 5 mM EGTA, 1% Triton X-100, 2 mM phenylmethylsulfonyl fluoride), homogenized, pelleted by centrifugation, and treated like total protein extracts (50). The following antibodies were used: Vinculin (1:1,000, Cat#ab88053; Abcam), Itga7 (1:1,000, Cat#Ab24509; Abcam), Talin (1:1,000, Cat#AHP1272; Bio-Rad), α-actinin (1:1,000, Cat#22170-1-AP; Proteintech), Chkβ (1: 250, Cat#398957; Santa Cruz), GAPDH (1:1,000, Cat#398957; Cell signaling). Proteins were visualized with goat anti-rabbit IRDye-800- or IRDye-680-secondary antibodies (LI-COR Biosciences) or anti-mouse m-IgGκ BP-CFL 790 (Cat. no.sc-516181; Santa Cruz) using an Odyssey imaging system and band density were evaluated using FIJI (NIH).

## Immunostaining and quantification of total and CSK-resistant vinculin

Quadriceps and gastrocnemius muscles were embedded in Optimal Cutting Temperature (Sakura Finetek), and were frozen in cooled isopentane in liquid nitrogen and stored at −80°C. Frozen sections (5 µm thick) were thaw-mounted on SuperFrost Microscope slides (Microm International) and air dried. For total vinculin, tissue sections were then fixed in 4% (wt/vol) PFA for 20 min. For CSK resistant vinculin, the frozen sectioned were first treated with CSK buffer (0.5% Triton X-100, 10 mM PIPES pH 6.8, 50 mM NaCl, 3 mM MgCl$_2$, 300 mM sucrose and complete protease inhibitor cocktail [Roche]) at 4°C for 1 min, followed by fixation with 4% PFA for 20 min. All \sections were blocked for 1 h with 10% donkey normal serum in PBS, followed by incubation with the primary antibodies overnight at 4°C. Slides were washed (3 × 5 min) in PBS and incubated for 1 h in the dark with 1:750 dilutions of the appropriate secondary antibodies (Donkey anti-goat or Rabbit) coupled with Alexa Fluor 594 or 488 (Molecular Probes) in PBS. Slides were washed five times in PBS, mounted with ProLong Gold Antifade (Molecular Probes), and were observed under a laser scanning confocal microscope (Zeiss LSM 710). Images were converted to 8-bit and the total corrected cellular fluorescence for the green channel was measured. A total of 20 random myofibers were quantified per group in three distinct mice per group using FIJI (NIH) software. The total corrected cellular fluorescence = integrated density − (area of selected cell x mean

---

inhibited cells than in vehicle-treated cells. **(C)** Quantification of PH-PLCD1-GFP fluorescence intensity at the plasma membrane in vehicle and choline kinase inhibited cells. Treatment with the choline kinase alpha inhibitors EB-3D and CK-37 resulted in reduced PH-PLCD1-GFP fluorescence intensity at the plasma membrane. Data show the mean ± SD (n = 10–18 cells per experiment, three independent experiments); (one-way ANOVA followed by Tukey's Multiple Comparison Test; ***$P < 0.001$). **(D)** Choline kinase inhibition decreases focal adhesion number in U2OS cells. U2OS cells were transiently transfected with GFP-vinculin and were treated with choline kinase alpha inhibitors EB-3D (10 mM), CK-37(15 mM) or vehicle for 48 h and imaged by spinning-disk confocal microscopy. Scale bars: 10 µm. Treatment with choline kinase alpha inhibitors significantly reduced focal adhesion numbers in U2OS cells compared with vehicle-treated cells. **(E)** The ratio of focal adhesion:cytoplasmic GFP-vinculin fluorescence is significantly decreased in choline kinase inhibited cells than in vehicle-treated cells. Data show the mean ± SD (n = 10–18 cells per experiment, three independent experiments); (one-way ANOVA followed by Tukey's Multiple Comparison Test; ***$P < 0.001$). **(F)** Overexpression of Myc-$CHKB$ in U2OS cells by lentivirus-mediated gene expression system. 24 h after lentiviral transfection, Myc-CHKB protein expression was determined by Western blot analysis. GAPDH was used as an internal control. **(G)** Choline kinase alpha inhibition by EB-3D (10 mM) or CK-37(15 mM) for 48 h did not reduce the plasma membrane localization of PH-PLCD1-GFP clusters in U2OS cells overexpressing Myc-CHKB. **(H)** The ratio of dorsal plasma membrane: cytoplasmic PH-PLCD1-GFP fluorescence in vehicle and choline kinase alpha inhibited cells overexpressing Myc-CHKB. **(I)** PH-PLCD1-GFP fluorescence intensity at the plasma membrane in vehicle and choline kinase alpha inhibited cells overexpressing Myc-CHKB. Data show the mean ± SD (n = 10–18 cells per experiment, three independent experiments); (one-way ANOVA followed by Tukey's Multiple Comparison Test; ***$P < 0.001$). **(J)** Choline kinase alpha inhibition by EB-3D (10 mM) or CK-37(15 mM) for 48 h was essentially without effect on focal adhesion numbers in U2OS cells transfected with Myc-CHKB. **(K)** The ratio of focal adhesion:cytoplasmic GFP-vinculin fluorescence is similar in vehicle or choline kinase inhibited cells. Data show the mean ± SD (n = 10 cells per experiment, three independent experiments); (one-way ANOVA followed by Tukey's Multiple Comparison Test; ns, not significant). Scale bar = 15 µm. Two additional independent experiments produced the same results.

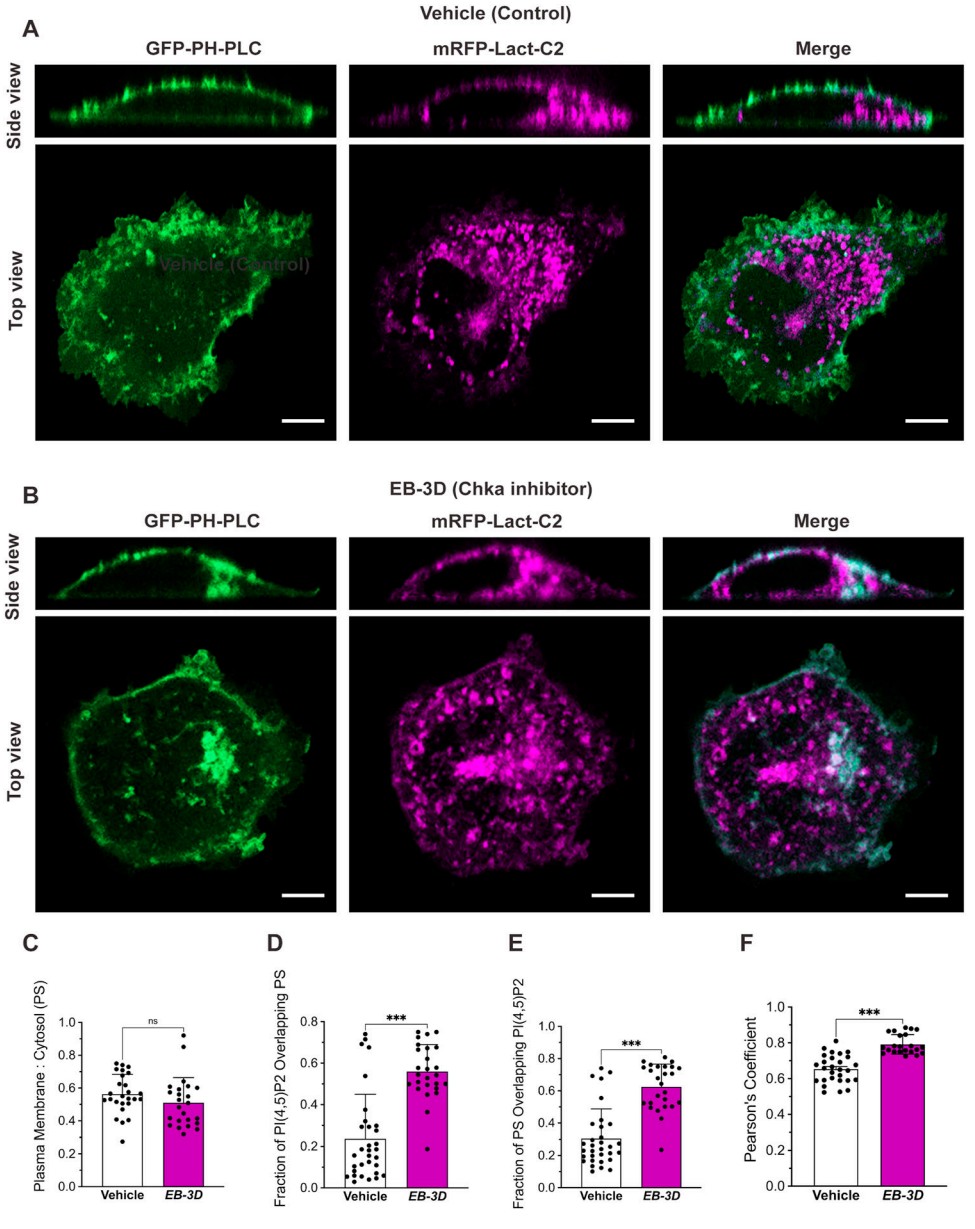

**Figure 5. Inhibition of choline kinase activity does not alter PS distribution with internalized PI(4,5)P$_2$ associating with enriched PS structures in U2OS cells.**
**(A, B)** U2OS cells were transiently co-transfected with the PI(4,5)P$_2$ reporter PH-PLCD1-GFP and PS reporter mRFP-Lact-C2 and were treated with DMSO (vehicle control) or the choline kinase alpha inhibitor EB-3D (15 mM) for 48 h and imaged by spinning-disk confocal microscopy in live cells. Choline kinase inhibition resulted in internalization of dorsal plasma membrane PH-PLCD1-GFP fluorescence where it colocalized PS containing organelles. Scale bar = 15 μm. **(C)** Quantification of the ratio of dorsal plasma membrane: cytoplasmic PS reporter mRFP-Lact-C2. **(D, E)** Quantification of the fraction of PI(4,5)P$_2$ overlapping PS (D), and the fraction of PS overlapping PI(4,5)P$_2$ (E) using the Manders' colocalization coefficients in vehicle-treated and choline kinase inhibited cells. **(F)** Colocalization between PI(4,5)P$_2$ and PS quantified as Pearson's correlation coefficient in vehicle-treated and choline kinase inhibited cells. For (C, D, E, F) data show the mean ± SD. N = 25 random cells per group. $*P < 0.05$, $**P < 0.01$, $***P < 0.001$ (t test).

fluorescence of background readings), was calculated and compared between groups.

### Plasmid constructs

PH-PLCD1-GFP was a gift from Tamas Balla (plasmid#51407; Addgene; http://n2t.net/addgene:51407; RRID:Addgene_51407) and is reported previously (42, 51). GFP-mouse vinculin full length (889) was a gift from Alpha Yap (plasmid #67935; Addgene; http://n2t.net/addgene:67935; RRID:Addgene_67935), mRFP-Lact-C2 was a gift from Sergio Grinstein (plasmid #74061; Addgene; http://n2t.net/addgene:74061; RRID:Addgene_74061) and was previously described (52). U2OS cells were transfected with 2 μg of the PH-PLCD1-GFP and/or 2 μg of GFP-mouse vinculin and 1 μg mRFP-Lact-C2 plasmids using lipofectamin2000 (Cat#11668019; Thermo Fisher Scientific).

### Live cell imaging

U2OS cells that had been transfected with PH-PLCD1-GFP, GFP-vinculin, or mRFP-Lact-C2 and grown on glass cover slips were treated with choline kinase alpha inhibitors EB-3D (10 mM), CK-37(15 mM) or vehicle for 48 h, placed into the imaging chamber (37°C, 5% CO$_2$) of a Zeiss Axio Observer Z.1 Spinning Disk Confocal Microscope and image acquisition was performed with Zen Black software.

### Quantification of the plasma membrane to cytosol ratio of PH-PLCD1-GFP, mRFP-Lact-C2, and GFP-vinculin

Quantification of the plasma membrane, plasma membrane to cytosol ratio of GFP-PH-PLCD1 (Fig 4A and B), and mRFP-Lact-C2 (Fig 5C) was performed using FIJI (National Institutes of Health) as

described elsewhere ([38], [42]). For each cell, the green or red pixel density was determined in the *X-Z* optical sections (side view) using three uniform squares overlaid on cytosol, dorsal plasma membrane (cell apex and immediately left and right), and outside the cell (background). After subtraction of background, the mean density was determined for cytosol and plasma membrane and the ratio was determined. Quantification was performed for 10 cells in each of three independent experiments. Quantification of the Focal Adhesions, cytosol ratio of GFP-vinculin (Fig 4C–F), was performed using FIJI (National Institutes of Health). For each cell, the green pixel density was determined in the X-Z optical sections (side view) using three uniform squares overlaid on cytosol, basal plasma membrane, and outside the cell (background). After subtraction of background, the mean density was determined for cytosol and basal plasma membrane and the ratio was determined. Quantification was performed for 10 to 18 cells in each of three independent experiments.

### Quantification of colocalization between PH-PLCD1-GFP and mRFP-Lact-C2

U2OS cells that had been transfected with PH-PLCD1-GFP and mRFP-Lact-C2 were grown on glass cover slips and treated with choline kinase alpha inhibitors EB-3D (15 mM) or vehicle for 48 h and imaged using Zeiss Axio Observer Z.1 Spinning Disk Confocal Microscope. For each group, confocal images from 25 random cells were quantified for colocalization of $PI(4,5)P2$ and PS using an imageJ plugin called JaCoP ([53]).

### Lentivirus-mediated gene expression

Lenti ORF of human choline kinase beta (CHKB), Myc-DDK-tagged (Cat #RC210253L1; OriGene Technologies, Inc) was used with the Lenti-vpak Lentiviral Packaging Kit (Cat #TR30037; OriGene Technologies, Inc). To produce the virus, 293T cells were plated on 10-cm plates (Corning) at $6.5 \times 106$ cells per plate in a complete DMEM medium with 10% serum and allowed to adhere for 16 h. Transfections were performed as per manufacturer's instructions with 4.5 $\mu g$ of transfer vector plasmid, 18 $\mu g$ packaging mixture (Invitrogen) and 67.5 $\mu l$ of Lipofectamine 2000 in serum-free Opti-MEM medium (Invitrogen). 24 h after transfection the cells were washed with PBS; fresh complete DMEM with pyruvate and 10% FBS was added and the cells were incubated for an additional 24 h. Vector supernatants were collected 48 h after transfection and filtered through a 0.45 $\mu m$ syringe filter (Whatman). The collected media containing lentiviral particles was diluted one to 4 times in fresh DMEM media containing 10% FBS and used for U2OS transduction.

U2OS cells were transduced with scrambled or Myc-CHKB–expressing lentiviral particles in the presence of polybrene (final concentration 8 $\mu g/ml$) and after 24 h Myc-CHKB expression was confirmed by Western blot analysis. Control and Myc-CHKB expressing U2OS cells were transiently transfected with 2 $\mu g$ of the GFP-vinculin using lipofectamin2000 (Cat#11668019; Thermo Fisher Scientific) and were treated with the choline kinase inhibitors EB-3D (10 mM), CK-37(15 mM) or vehicle for 48 h and imaged by spinning-disk confocal microscopy in live cells.

### Quantification and statistical analysis

All experiments were repeated three or more times. Data are presented as mean ± SEM or mean ± SD, as appropriate. For comparison of two groups the two-tailed *t* test was used unless otherwise specified. Comparison of more than two groups was performed by one-way ANOVA followed by the Tukey's Multiple Comparison test. *P*-values < 0.05 were considered significant.

## Supplementary Information

## Acknowledgements

This work was supported by a grant from the Canadian Institutes of Health Research (SOP-159230) to CR McMaster. The authors declare no competing financial interests.

### Author Contributions

M Tavasoli: formal analysis, validation, investigation, methodology, and writing—original draft, review, and editing.
CR McMaster: conceptualization, formal analysis, supervision, funding acquisition, project administration, and writing—original draft, review, and editing.

### Conflict of Interest Statement

The authors declare that they have no conflict of interest.

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
