## [Reviewer comments · Life Science Alliance]

Life Science Alliance

Defects in Integrin Complex Formation Promote CHKB Mediated Muscular Dystrophy

Mahtab Tavasoli and Christopher McMaster
DOI: <https://doi.org/10.26508/lsa.202301956>

Corresponding author(s): Christopher McMaster, Dalhousie University and Mahtab Tavasoli, Dalhousie University

Review Timeline:

Submission Date:	2023-01-27
Editorial Decision:	2023-02-27
Revision Received:	2023-12-15
Editorial Decision:	2024-01-04
Revision Received:	2024-05-01
Editorial Decision:	2024-05-02
Revision Received:	2024-05-03
Accepted:	2024-05-07

Transaction Report:

February 27, 2023

Re: Life Science Alliance manuscript #LSA-2023-01956-T

Dr. Christopher R McMaster
Dalhousie University
Pediatrics and Biochemistry & Molecular Biology
Atlantic Research Centre
5849 University Avenue, Rm C302
Halifax, NS B3H 4H7
Canada

Dear Dr. McMaster,

Thank you for submitting your manuscript entitled "Defects in Integrin Complex Formation Promote CHKB Mediated Muscular Dystrophy" to Life Science Alliance. The manuscript was assessed by expert reviewers, whose comments are appended to this letter. We invite you to submit a revised manuscript addressing the Reviewer comments.

Thank you for this interesting contribution to Life Science Alliance. We are looking forward to receiving your revised manuscript.

Sincerely,

Eric Sawey, PhD
Executive Editor
Life Science Alliance
<http://www.lsa-journal.org>

B. MANUSCRIPT ORGANIZATION AND FORMATTING:

Reviewer #1 (Comments to the Authors (Required)):

CHKB is required for PC synthesis and linked to muscular dystrophy related loss of membrane integrity. The effects are known to affect the attachment of integrin-vinculin-talin. PIP2 has been shown to be involved in the recruitment of these proteins to PM. The authors have basically reconfirmed these previously published data. They show that in the absence of CHKB, vinculin is dissociated from the membrane. They also show that inhibitors of PIP2 affect recruitment of integrin protein complex to the membrane. The authors need to go beyond this yes/no kind of observation and make this study a bit more molecular and quantitative in analysis. How does ChkB affect PIP2 levels? Is that the only defect or is there another lipid molecule involved in the recruitment of the proteins mentioned above. What part of the protein (s) bind PIP2? In all, there is nothing wrong with the data, but it does not really add anything new.

Reviewer #2 (Comments to the Authors (Required)):

Certain congenital forms of muscular dystrophy are due to autosomal recessive loss of function mutations in the CHKB gene (e.g. see Nishino et al. 1998; Mitsuhashi et al. 2011). CHKB encodes the choline kinase beta enzyme that drives the first step in the Kennedy pathway for the synthesis of phosphatidylcholine (PC), the most abundant phospholipid in eukaryotic cell membranes including the sarcolemma of muscle cells. Previous studies have reported that impaired choline kinase beta (CHKB) function and PC metabolism lead to mitochondrial dysfunction as a key step in the progression of muscular dystrophy (Nishino et al. 1998; Mitsuhashi et al. 2011). While previous studies linking CHKB, lipid metabolism, and mitochondrial dysfunction are certainly informative, the current study by Tavasoli and McMaster reveals potentially new and important insight into how defects in CHKB-dependent PC synthesis leads to muscular dystrophy. In particular, the data uncover defects in integrin-mediated focal adhesions in muscle cells from CHKB^{-/-} mice. Paradoxically, steady-state levels of the phospholipid PC are only modestly changed in muscle cells from CHKB^{-/-} mice, making it unclear as to what lipids may be significantly affected. This study suggests that alterations in phosphatidylinositol 4,5-bisphosphate (PtdIns4,5P2) metabolism may ultimately impact focal adhesion stability. Thus, modest derangements in CHKB-dependent PC metabolism appear to be a root cause in the progression of muscular dystrophy by conferring defects in both mitochondrial and focal adhesion structure and function.

However, prior to publication, a few additional experiments are recommended to better understand how alterations in CHKB-dependent PC synthesis impact PtdIns4,5P2 metabolism and focal adhesion stability. The effects on vinculin-, talin-, and integrin-mediated focal adhesions (costameres) appear robust and striking in muscle cells from CHKB^{-/-} mice (Figures 1, 2, and 3). My specific comments mainly involve the experiments using U2O2 cells (Figure 4), addressing how slight alterations in PC metabolism might result in significant effects upon PtdIns4,5P2 localization and focal adhesion stability.

Major comments:

1. Because anionic lipids including phosphatidic acid (PA) and phosphatidylserine (PS) stimulate PIP 5-kinase (PIP5K) activities that generate PtdIns4,5P2 at the plasma membrane (Fairn et al 2009), and because PS synthase generates PS from PC, and because phospholipase D (PLD) generates PA from PC, the authors should examine the localization of PS and PA reporters (e.g. using LactC2 for PS and Spo20 or Q2 for PA) upon CHKB inhibition.
2. Along these lines, the authors may want to examine whether expression of hyperactive PS synthase or over-expression of a PLD isoform rescues any phenotypes in U2O2 cells conferred by CHKB inhibition.
3. The authors imply that PtdIns4,5P2 is 'internalized' upon CHKB inhibition (see the Abstract). However, it has been demonstrated that PIP5Ks mis-localize to intracellular compartments upon loss of certain lipids (such as PS and PA) at the plasma membrane, resulting in the generation of PtdIns4,5P2 at intracellular PtdIns4P-containing compartments (Nakatsu et al 2012). The authors should examine the localization of PIP5K isoforms upon CHKB inhibition. Does re-localization of PIP5K to intracellular compartments cause the mis-localization of the PtdIns4,5P2 reporter to intracellular compartments? Alternatively, to test if PtdIns4,5P2 is indeed 'internalized' upon CHKB inhibition, the authors may want to test whether inhibition of endocytosis,

such as by using the dynamin inhibitor Dynasore, prevents PtdIns4,5P2 'internalization'.

4. Figures 4E and 4F show that overexpression of myc-tagged CHKB rescues the vinculin phenotypes upon endogenous CHKB inhibition using the small molecule compounds EB-3D and CK-37. These results provide an important control demonstrating that the small molecule compounds are likely not exerting indirect off-target effects. However, important additional control tests are missing from this experiment. First, it would be useful to confirm myc-CHKB expression in the cells examined. More importantly, does overexpression of myc-CHKB restore proper localization of the PtdIns4,5P2 reporter to the plasma membrane? If not, is rescue of vinculin plasma membrane localization due to some other reason?

Minor comments:

1. Use of the abbreviation "PIP2" is not appropriate as it could refer to several phosphoinositide isoforms including PtdIns4,5P2, PtdIns3,5P2, and PtdIns3,4P2. In the Abstract, the authors appropriately define the abbreviation "PC" for phosphatidylcholine, but they do not define "PIP2" until later in the Introduction. Phosphatidylinositol 4,5-bisphosphate should be spelled out in the Abstract.

2. The minor effects on PC levels in CHKB^{-/-} cells seem very interesting (and even paradoxical) and the authors may want to speculate on this further in the Discussion. Perhaps I missed some key information relative to this issue that is provided in the manuscript, but what is known about potential CHKA upregulation, or increased flux through the PS decarboxylase-dependent PC pathway, or most importantly whether small changes in PC are indeed known to affect PS or PA metabolism?

Response to Reviewers' comments (LSA-2023-01956-T)

Reviewer 1

Bi-allelic loss of function variants in *CHKB*, encoding the first step in the synthesis of PC, is the cause of a rostrocaudal muscular dystrophy in both humans and mice. Loss of sarcolemma integrity is a hallmark of muscular dystrophies, however, how this occurs in the absence of choline kinase function is not known. We had previously determined that in affected muscle in *Chkb*^{-/-} mice, versus unaffected muscle, there is a failure of the $\alpha7\beta1$ integrin complex that is specific to affected muscle. As the major functional components of the $\alpha7\beta1$ integrin complex need to bind to the anionic lipid PI(4,5)P₂ on the cytoplasmic facing leaflet of the plasma membrane to be functional, and we had determined that choline kinase inhibition results in redistribution of PI(4,5)P₂ from the plasma membrane to internal structures. This provides an explanation for the loss of plasma membrane integrity specific to affected muscle in *Chkb* mediated muscular dystrophy. In the revised manuscript we went on to determine if this redistribution of PI(4,5)P₂ was specific to PI(4,5)P₂. To do so, we interrogated the distribution of the major cytoplasm facing anionic lipid PS upon inhibition of choline kinase activity. PS distribution did not change upon choline kinase inhibition indicating the plasma membrane itself is not affected in a large overt manner, and that the loss of plasma membrane integrity is due to internalization of PI(4,5)P₂ from the plasma membrane resulting in dissolution of the $\alpha7\beta1$ integrin complex.

Reviewer 2

Major comments

1-4. We worked with Dr. Gregory Fairn, an expert in using probes to determine lipid localization to aid in addressing the reviewer's concerns. Fortunately, Dr Fairn was a former PhD student of mine and was recently recruited from the University of Toronto to Dalhousie University and his office is now next to mine. As suggested, we determined the localization of PS in U2O2 cells upon choline kinase inhibition using the mRFP-LactC2 probe. We also looked at PA localization using a Spo20 probe, however, the PA signal was too low for accurate PA localization determination.

Like PI(4,5)P₂, PS is an anionic lipid that is present on the cytoplasmic leaflet of the PM. Unlike PI(4,5)P₂ which is normally highly localized to the PM, PS is also present in late endocytic compartments and recycling endosomes. Our new experiments in the revised manuscript have shown that PS localization was unchanged when choline kinase was inhibited suggesting that (i) PS localization is not dependent on PC synthesis via the Kennedy pathway and (ii) there are no large overt changes in the PM upon choline kinase inhibition. Interestingly, PI(4,5)P₂ that was internalized colocalized with the internal PS pool suggesting that PI(4,5)P₂ may be present in endosomes upon internalization. As PS was not lost from the PM, we felt the suggestions to determine the localization of enzyme that were dependent on PS would not add to the manuscript.

Minor comments

1. Thank you for noting that PIP2 is not specific to PI(4,5)P₂. We have used PI(4,5)P₂ throughout the manuscript.

2. We previously used lipidomics to interrogate changes in the levels and species of the major cellular lipids in affected and unaffected muscle from *Chkb*^{+/+}, *Chkb*^{+/-}, and *Chkb*^{-/-} mice over time, as well as in myocytes isolated from these same mice. There was no indication of changes in flux through any of the PC synthesis pathways Mechanism of action and therapeutic route for a muscular dystrophy caused by a genetic defect in lipid metabolism, M. Tavasoli, S. Lahire, S. Sokolenko, R. Novorolsky, S. A. Reid, A. Lefsay, et al., Nat Commun 2022 Vol. 13 Issue 1 Pages 1559, instead, it has been previously demonstrated that the level of PC is maintained in affected muscle in *Chkb*^{-/-} mice via an increase in PC uptake from serum (Understanding the muscular dystrophy caused by deletion of choline kinase beta in mice, G. Wu, R. B. Sher, G. A. Cox and D. E. Vance, Biochim Biophys Acta 2009 Vol. 1791 Issue 5 Pages 347-56). We have added a statement to this affect in the Introduction (page 4, highlighted in yellow). Thank you for pointing out this important point that required clarification.

January 4, 2024

Re: Life Science Alliance manuscript #LSA-2023-01956-TR

Dr. Christopher R McMaster
Dalhousie University
Pharmacology
5850 College St
Room 1-A1
Halifax, NS B3H 4H7
Canada

Dear Dr. McMaster,

Thank you for submitting your revised manuscript entitled "Defects in Integrin Complex Formation Promote CHKB Mediated Muscular Dystrophy" to Life Science Alliance. The manuscript has been seen by one of the original reviewers whose comments are appended below, and some important issues remain.

Our general policy is that papers are considered through only one revision cycle; however, we are open to one additional short round of revision. Please note that I will expect to make a final decision without additional reviewer input upon re-submission.

Please submit the final revision within one month, along with a letter that includes a point by point response to the remaining reviewer comments.

To upload the revised version of your manuscript, please log in to your account: <https://lsa.msubmit.net/cgi-bin/main.plex>
You will be guided to complete the submission of your revised manuscript and to fill in all necessary information.

-- A letter addressing the reviewer comments point by point.

B. MANUSCRIPT ORGANIZATION AND FORMATTING:

Sincerely,

Reviewer #2 (Comments to the Authors (Required)):

In their revised manuscript, Tavasoli and McMaster maintain that a deficiency in CHKB-dependent PC metabolism results in

PtdIns4,5P2 'internalization', resulting in a loss of integrin complexes. This finding is potentially important and may provide new insight into how CHKB deficiency may ultimately lead to muscular dystrophy.

A major finding of this study is that a PtdIns4,5P2 biosensor localizes to some unidentified intracellular compartments upon impaired CHKB-dependent PC metabolism. Both reviewers thought it would be important to determine why the PtdIns4,5P2 biosensor becomes mis-localized. This issue was not addressed in the revised manuscript, which may require a great deal of work and perhaps the authors plan to address this in detail in a future study.

However, because this issue was not addressed in the current study, some of the authors conclusions remain untested and unsubstantiated. First, the authors conclude that CHKB inhibition results in "internalization" of a PtdIns4,5P2 reporter (see the Abstract). This wording could be interpreted to suggest that PtdIns4,5P2 is endocytosed and accumulates on endosomal compartments due to impaired 5-phosphatase activity in CHKB deficient cells. In this case, blocking endocytosis by the dynamin inhibitor Dynasore may prevent PtdIns4,5P2 'internalization' upon CHKB inhibition. Yet, this possibility was not addressed in the revised manuscript. Another possibility is that PIP5K isoforms mis-localize to intracellular compartments upon loss of CHKB function, resulting in the generation of PtdIns4,5P2 at intracellular compartments. The authors chose not to examine the localization of PIP5K isoforms upon CHKB inhibition in the revised manuscript. So, it remains unknown how the PtdIns4,5P2 reporter becomes mis-localized upon CHKB inhibition, and the authors' conclusion that PtdIns4,5P2 is 'internalized' has not been demonstrated. Given the lack of mechanistic data, perhaps the authors should simply state that the PtdIns4,5P2 reporter is mis-localized or re-distributed to intracellular compartments.

It is also unknown whether mis-localization of the PtdIns4,5P2 reported correlates with the observed effects on integrin complexes. The data show that overexpression of myc-tagged CHKB rescues vinculin localization upon inhibition of endogenous CHKB using small molecule compounds. It may be informative to address whether myc-CHKB overexpression also restores proper localization of the PtdIns4,5P2 reporter to the plasma membrane. If not, the effects observed on vinculin localization may be due to some other reason other than PtdIns4,5P2 re-distribution.

The revised manuscript also includes a model cartoon that may be a bit misleading. The cartoon suggests that PtdIns4,5P2 levels are decreased at the plasma membrane in CHKB deficient cells. However, there are no direct biochemical measurements of PtdIns4,5P2 levels or metabolism in this study. Likewise, there are no quantitative measurements of the PtdIns4,5P2 reporter levels at the plasma membrane. Perhaps the authors could perform ratiometric analyses of the PtdIns4,5P2 and PS reporters at the plasma membrane to address this possibility, since the data are already obtained but not yet quantitated. Measurements of plasma membrane:cytosol ratios of the PtdIns4,5P2 reporter do not necessarily indicate reduced levels of the PtdIns4,5P2 at the plasma membrane. Instead, the only conclusion that can be made from the results in the current study is that the PtdIns4,5P2 reporter is re-distributed to intracellular compartments in CHKB deficient cells, and the model cartoon does not even depict this finding. The cartoon is also missing a key interaction between talin and PtdIns4,5P2 at the plasma membrane.

In summary, because important issues remain unaddressed, some of the authors' conclusions (as stated in the manuscript text and implied in the model cartoon) are overstated. Some revisions to the manuscript and model are necessary to more accurately describe the findings of this study, and the authors may even wish to perform quantitative analyses of the PtdIns4,5P2 reporter at the plasma membrane (using data already in hand) to provide some support of their conclusions depicted in the model.

Response to Reviewers' comments (LSA-2023-01956-TR)

Thank you for pointing out that our explanation for redistribution of the PI(4,5)P₂ probe does not necessarily mean that PI(4,5)P₂ itself has been internalized by any means, including endocytosis. Indeed, the data could also point to the fact that there is less of the PI(4,5)P₂ probe at the plasma membrane due to decreased synthesis or increased catabolism of PI(4,5)P₂ itself. Hence, as requested by the reviewer and have modified the language throughout the manuscript from PI(4,5)P₂ was internalized, to the PI(4,5)P₂ probe was redistributed, to more accurately reflect the data to more accurately describe our findings.

The reviewer suggested an experiment where endocytosis be blocked using the dynamin inhibitor Dynasore and assess PI(4,5)P₂ probe localization upon simultaneous pharmacological inhibition of choline kinase activity. These experiments were performed. Unfortunately, we found that when choline kinase was inhibited the addition of Dynasore to also inhibit endocytosis (at a plethora of concentrations and at various time points after choline kinase inhibition) resulted in rapid cell death.

We performed the requested experiment whereby we determined the localization of the PI(4,5)P₂ probe upon addition of the choline kinase inhibitor along with simultaneous over-expression of CHKB. In the previous version we had shown that this resulted in the redistribution of vinculin back to the plasma membrane upon over-expression of CHKB, and the new data is consistent with this observation as we now show that that the PI(4,5)P₂ probe also redistributes back to the plasma membrane over-expression of CHKB (Fig 4 G, H, and I).

We agree that the cartoon model could be misleading. Although we demonstrate that the PI(4,5)P₂ probe does redistribute from the plasma membrane to the cytosol due to choline kinase inhibition, and that once in the cytosol the PI(4,5)P₂ probe does overlap with the PS probe (which is known to be in endosomes), we cannot rule out that PI(4,5)P₂ metabolism could also be altered as a means of reducing PI(4,5)P₂ plasma membrane content. For the sake of not “muddying the waters” we have omitted the model.

May 2, 2024

RE: Life Science Alliance Manuscript #LSA-2023-01956-TRR

Dr. Christopher R McMaster
Dalhousie University
Pharmacology
5850 College St
Room 1-A1
Halifax, NS B3H 4H7
Canada

Dear Dr. McMaster,

Thank you for submitting your revised manuscript entitled "Defects in Integrin Complex Formation Promote CHKB Mediated Muscular Dystrophy". We would be happy to publish your paper in Life Science Alliance pending final revisions necessary to meet our formatting guidelines.

- please be sure that the authorship listing and order is correct
- please add ORCID ID for the secondary corresponding author--they should have received instructions on how to do so
- please create a separate Author Contributions section of your main manuscript text
- please create a separate conflict of interest statement of your main manuscript text
- please use the [10 author names et al.] format in your references (i.e., limit the author names to the first 10)
- in the Materials and Methods section, please state that approval for animal use was granted, and who provided this approval

FIGURE CHECKS

- we encourage you to arrange Figure 1 so that the panels appear in alphabetical order, and update the legend and text callouts accordingly
- please add callouts for Figure 4G, H and I

A. FINAL FILES:

B. MANUSCRIPT ORGANIZATION AND FORMATTING:

Sincerely,

May 7, 2024

RE: Life Science Alliance Manuscript #LSA-2023-01956-TRRR

Dr. Christopher R McMaster
Dalhousie University
Pharmacology
5850 College St
Room 1-A1
Halifax, NS B3H 4H7
Canada

Dear Dr. McMaster,

Thank you for submitting your Research Article entitled "Defects in Integrin Complex Formation Promote CHKB Mediated Muscular Dystrophy". It is a pleasure to let you know that your manuscript is now accepted for publication in Life Science Alliance. Congratulations on this interesting work.

DISTRIBUTION OF MATERIALS:

Again, congratulations on a very nice paper. I hope you found the review process to be constructive and are pleased with how the manuscript was handled editorially. We look forward to future exciting submissions from your lab.

Sincerely,
